# Targeting Interleukin-13 Receptor α2 and EphA2 in Aggressive Breast Cancer Subtypes with Special References to Chimeric Antigen Receptor T-Cell Therapy

**DOI:** 10.3390/ijms25073780

**Published:** 2024-03-28

**Authors:** Dharambir Kashyap, Huda Salman

**Affiliations:** Brown Center for Immunotherapy, Melvin and Bren Simon Comprehensive Cancer Center, Division of Hematology and Oncology, School of Medicine, Indiana University, Indianapolis, IN 46202, USA; dbir@iu.edu

**Keywords:** triple-negative breast cancer, interleukin-13 receptor α2, EphA2, immunotherapy

## Abstract

Breast cancer (BCA) remains the leading cause of cancer-related mortality among women worldwide. This review delves into the therapeutic challenges of BCA, emphasizing the roles of interleukin-13 receptor α2 (IL-13Rα2) and erythropoietin-producing hepatocellular receptor A2 (EphA2) in tumor progression and resistance. Highlighting their overexpression in BCA, particularly in aggressive subtypes, such as Her-2-enriched and triple-negative breast cancer (TNBC), we discuss the potential of these receptors as targets for chimeric antigen receptor T-cell (CAR-T) therapies. We examine the structural and functional roles of IL-13Rα2 and EphA2, their pathological significance in BCA, and the promising therapeutic avenues their targeting presents. With an in-depth analysis of current immunotherapeutic strategies, including the limitations of existing treatments and the potential of dual antigen-targeting CAR T-cell therapies, this review aims to summarize potential future novel, more effective therapeutic interventions for BCA. Through a thorough examination of preclinical and clinical studies, it underlines the urgent need for targeted therapies in combating the high mortality rates associated with Her-2-enriched and TNBC subtypes and discusses the potential role of IL-13Rα2 and EphA2 as promising candidates for the development of CAR T-cell therapies.

## 1. Introduction

Breast cancer (BCA) represents the foremost cause of cancer-related mortality after cardiovascular disease among women, accounting for 31% (297,790 cases) of all cancer diagnoses in 2023, followed by lung/bronchus (13%) and colorectal (8%) [1]. The challenges in BCA treatment include therapeutic resistance, the emergence of acquired resistance, and the absence of precise, targetable biomarkers with consistent expression on cancer cells. The interleukin-13 receptor α2 (IL-13Rα2) and the erythropoietin-producing hepatocellular receptor A2 (EphA2) have been identified as playing crucial roles in tumor growth and progression [2,3]. EphA2, a constituent of the Eph kinase subfamily, falls within the receptor tyrosine kinase (RTK) superfamily [4]. Ephrin A1 serves as the primary ligand for EphA2, facilitating cell–cell interactions necessary for the transmission of bidirectional signals between adjacent cells [5]. In cells expressing EphA2, a forward signal is relayed, whereas in those expressing ephrin A1, a reverse signal is conveyed. As such, EphA2 is pivotal for intercellular communication, influencing both normal physiological processes and pathological conditions, including cancer [6,7]. The signaling pathway involving EphA2 and ephrin A1 is aberrantly regulated in various tumor entities, particularly in breast cancer [8,9]. Elevated expression of EphA2 in BCA is associated with tumor progression and an adverse prognosis [4,10]. However, the specific mechanisms through which EphA2 fosters BCA progression remain to be fully elucidated. IL-13Rα2 is a membrane-associated protein encoded by the IL-13RA2 gene [11]. Extensive research has identified IL-13Rα2 overexpression not only in glioblastoma multiforme (GBM) but also in advanced stages of BCA, where it correlates with poor prognosis and enhanced metastatic capability [3,12,13]. The anti-inflammatory Th2 cytokine IL-13 exhibits a high affinity for the IL-13Rα2 receptor. Studies have elucidated that IL-13Rα2 expression and subsequent IL-13Rα2-mediated signaling pathways contribute to tumor proliferation, invasion, metastasis, and cellular survival across a spectrum of human cancers, including GBM, prostate, lung, ovarian, pancreatic, gastric, and breast carcinomas [14,15,16,17,18,19,20]. Tumor cells manipulate IL-13Rα2 receptor expression to support tumorigenesis, progression, and immune evasion mechanisms [17,18,19,20]. Both IL-13Rα2 and EphA2 receptors are expressed in BCA, displaying differential expression patterns between normal and malignant tissues, making them promising targets for therapeutic interventions [3,21]. The U.S. Food and Drug Administration (FDA) has sanctioned numerous immunotherapeutic agents, encompassing monoclonal antibodies and chimeric antigen receptor T-cell (CAR-T) therapies, for a variety of cancers [22]. Given their aberrant overexpression in cancerous tissues relative to normal counterparts, IL-13Rα2 and EphA2 receptors emerge as potential immunotherapeutic targets in BCA. CAR T-cell therapy, developed over the past decade, encompasses several generations of CAR T-cells tailored for hematologic malignancy treatment [22,23]. Ongoing clinical trials are investigating CAR T-cell efficacy in solid tumors, including BCA, with promising outcomes in various phases of clinical research. This review article focuses on the therapeutic potential of IL-13Rα2 and EphA2 as biomarkers for the development of CAR T-cell therapy targeting more aggressive subtypes of BCA.

## 2. Structure and Function of IL-13Rα2 and EphA2

EphA2 is classified within the RTK superfamily, characterized as a 130 kDa glycoprotein comprising 967 amino acids, and functions as a type I transmembrane glycoprotein. The *EPHA2* gene, responsible for encoding EphA2, is situated on chromosome 1p36.13 [24]. Structurally, the extracellular domain of EphA2 encompasses a ligand-binding domain (LBD), a Sushi domain, and an epidermal growth factor-like (EGF) domain, along with two fibronectin type III domains (FN1 and FN2) (Figure 1). The intracellular region of EphA2 is composed of a tyrosine kinase domain, a sterile α-motif (SAM) domain, and a PDZ-binding motif, all of which are linked to the extracellular domain via a singular transmembrane (TM) helix, bordered by juxtamembrane linkers. The C-terminal segment of EphA2 serves as an anchorage point for the association of various proteins, including PSD-95 (postsynaptic density protein), discs large (DLG), and zonula occludens-1 (ZO-1), facilitating interactions with postsynaptic density components, synaptic scaffolding molecules, and tight junction proteins, respectively [5,7,25]. The juxtamembrane domain of EphA2 features two pivotal autophosphorylation sites, tyrosines Y588 and Y594, which are essential for its interaction with guanine nucleotide exchange factors Vav2 and Vav3, thereby playing a crucial role in signal transduction processes [7,25].

IL-13Rα2 is a glycosylated transmembrane protein composed of 380 amino acids, with a molecular weight of approximately 56 kDa, encoded by the *IL13RA2* gene located on the X chromosome within regions Xq13.1-q28 [26]. The protein’s structure includes a substantial extracellular domain consisting of 317 amino acids, a transmembrane domain of 20 amino acids, a lengthy N-terminal signal sequence of 26 amino acids, and an intracellular domain of 17 amino acids, which notably lacks any signaling motifs. The extracellular domain features a “cytokine-binding homology region” (CHR), which encompasses a WXSWS motif, two fibronectin III-like segments (D2 and D3 loops), and an N-terminal S type-Ig region (D1 loop), facilitating cytokine binding [26,27] (Figure 1). Due to the absence of signaling motifs within its cytoplasmic tail, IL-13Rα2 is often referred to as a “decoy receptor”, binding to IL-13 with a significantly higher affinity—approximately 50 times greater—than that of IL-13Ra1. IL-13Rα2 possesses both IL-13-dependent and IL-13-independent functionalities that are capable of transducing signals through AP-1 family members FRA-2 and c-JUN, subsequently activating the transforming growth factor-ß1 (TGF-β1) promoter and inducing fibrosis [27]. Additionally, IL-13Rα2’s interaction with focal adhesion kinase (FAK) and phosphoinositide 3-kinase (PI3K) pathways necessitates the scaffold protein FAM120A (family with sequence similarity 120 member A), particularly during colon cancer metastasis. Studies have identified elevated levels of IL-13Rα2 in BCA cases that have metastasized to the lungs [28,29].

## 3. IL-13Rα2 and EphA2 in Her-2-Enriched and TNBC Pathology

EphA2 overexpression is observed in Her-2-enriched and triple-negative breast cancer (TNBC), the more aggressive subtypes of BCA, correlating with adverse prognostic outcomes. Research conducted in 2016 on Her-2-enriched tumors revealed that the ephrin A1/EphA2 signaling axis modulates glutamine metabolism [30]. Immunohistochemical analysis of 134 invasive BCA specimens demonstrated elevated EphA2 expression in 110 cases compared to normal breast tissue (ASCO Annual Meeting Proceedings, 2005). Notably, patients exhibiting EphA2 positivity were associated with shorter relapse-free survival (RFS) and overall survival (OS) [31,32] (Figure 2a). Similarly, TNBC patients with brief RFS periods showed heightened EphA2 expression [10]. Data from The Cancer Genome Atlas (TCGA) indicated a positive correlation between EphA2 expression and lymph node metastasis, as well as the Nottingham prognostic index in TNBC, contrasting with estrogen receptor-positive tumors [8]. Further studies have suggested the potential of EphA2 suppression to increase TNBC cell sensitivity, highlighting EphA2 as a viable therapeutic target [33]. In vitro experiments demonstrated that EphA2 loss of function significantly curtailed MDA-MB-231 cell proliferation and induced G1/S cell cycle phase arrest [34]. Mechanistically, EphA2 has been shown to regulate the CDK2/Cyclin E1/2 complex via the proteasome-mediated degradation of p27^KIP1, thereby inhibiting cancer cell growth [34,35]. Additionally, tropomyosin-related kinase A (TrkA) recruits EphA-2, which subsequently enhances BCA cell invasion; conversely, TrkA inhibition reverses cancer cell migration [36,37]. Overexpressed EphA2 has been implicated in resistance to Trastuzumab in Her-2-enriched BCA cells (SK-BR-3 and BT-474) through the increased phosphorylation of Src at Y416 and the activation of EphA2-PI3K/AKT and EphA2-Ras/MAPK signaling pathways [38] (Figure 3). Loss of function experiments involving EphA2 demonstrated a substantial reduction in tumor growth, angiogenesis, and lung metastasis in EphA2 knockout female mice [32]. Additionally, clinical samples from patients with basal-like BCA exhibiting phosphorylated EphA2 (pS897) were found to have the lowest response rates to neoadjuvant treatments and exhibited increased cancer stemness [39].

IL-13Rα2 expression has been implicated in the development of brain metastases in BCA patients. Investigations into TNBC and Her-2-enriched cancer models have revealed a correlation between IL-13Rα2 expression, the occurrence of brain metastases, and diminished survival outcomes [3] (Figure 2b). Microarray analyses of invasive BCA samples identified elevated IL-13Rα2 expression in Her-2-enriched and TNBC tumors, which was associated with poorer disease-free survival rates [40]. Compared to luminal primary BCA cells, basal cancer cells exhibited significantly higher levels of the IL-13Rα2 decoy receptor. Furthermore, patients with lower IL-13Rα2 expression levels were observed to have a more favorable prognosis and longer metastasis-free survival compared to those with higher expression levels [29].

## 4. Overcoming Therapeutic Hurdles: The Quest to Tame Aggressive Her-2-Positive and TNBC Breast Cancers

Gene expression profiling has delineated distinct molecular subtypes of BCA, utilizing estrogen receptor (ER), progesterone receptor (PR), and Her-2 as clinical biomarkers [41,42]. This classification stratifies BCA into various intrinsic subtypes through immunohistochemistry: Luminal A (ER and/or PR positive, Her-2 negative), Luminal B (ER and/or PR positive, Her-2 positive), Her-2 enriched (ER and PR negative, Her-2 positive), and TNBC (ER negative, PR negative, Her-2/neu negative) [43]. These subtypes exhibit unique gene expression profiles, dictating divergent clinical outcomes [44,45]. Her-2 positive and TNBC subtypes account for 20% and 25% of invasive breast cancers, respectively. These two subtypes are highly heterogeneous, especially TNBC, due to different course, prognosis, sensitivity to therapy, and variety of genetics. Patients with these two subtypes have poor outcomes, a tendency of brain involvement, higher rates of disease recurrence, and unfortunately, mortality. The Nottingham prognostic index (NPI) is a parameter to predict primary breast cancer survival based on tumor size, tumor grade, and lymph node involvement. Patients with Her-2 positive and TNBC usually showed moderate to poor NPI scores (NPI > 4.5) at the time of diagnosis.

Clinicopathological parameters, including receptor status, tumor dimensions, nodal involvement, metastatic spread, and patient age, inform therapeutic decisions and prognostication of recurrence risk [46,47]. High-throughput sequencing technologies underpinning phase III clinical trials have validated tools, such as Oncotype Dx and MammaPrint, for assessing recurrence and metastasis risk, findings that have been integrated into the National Comprehensive Cancer Network (NCCN) guidelines to refine treatment protocols for hormone-positive breast cancers. Her-2-enriched tumors are treated with the FDA-approved monoclonal antibody Trastuzumab, whereas luminal subtypes are managed with endocrine therapies, including Tamoxifen and Aromatase Inhibitors (for post-menopausal patients), targeting hormone receptors [48,49,50,51]. Conversely, TNBC patients, lacking ER, PR, and Her-2 expression, do not benefit from receptor-targeted therapies [52]. Comparative clinical analyses reveal that luminal subtype patients generally experience more favorable prognoses and overall survival compared to those with Her-2-positive and TNBC subtypes [44,45]. TNBC and Her-2-positive subtypes collectively constitute approximately 15–20% of all BCA cases [53,54]. Despite the availability of therapeutic alternatives, these subtypes are associated with the most adverse prognoses. Even with the implementation of adjuvant therapies, including chemotherapy and trastuzumab, around 20% of patients with Her-2-enriched BCA experience metastasis and recurrence [55]. Recent therapeutic advancements, such as tyrosine kinase inhibitors (TKIs), antibody–drug conjugates (ADCs), and novel monoclonal antibodies (mAbs), have been introduced for the aggressive treatment of Her-2/neu-positive patients. Nonetheless, a significant proportion of these patients develop treatment resistance within five years of follow-up [56,57]. Proposed mechanisms for acquired resistance include mutations in HER family genes (e.g., *HER2*L755S, *HER3*E928G), loss or masking of the HER2 epitope (p95HER2), activation of alternative signaling pathways (e.g., *PIK3CA* mutations, *PTEN* loss, cyclin D1-CDK4/6 axis), HER2 heterogeneity, and an immunosuppressive tumor microenvironment (TME) [58,59,60,61,62,63,64]. TNBC patients, in comparison, exhibit shorter survival periods and a 40% mortality rate within the first five years post-diagnosis [65]. TNBC is characterized by high metastatic potential and shorter survival times, and is more prevalent in younger women, correlating with a poorer prognosis [66,67,68,69]. Unlike other breast cancer subtypes, TNBC is often linked with genetic predispositions, including mutations in the *BRCA1* or BRCA2 genes [70]. Therapeutic options for TNBC patients are limited because it is the most immunogenic, aggressive, and heterogeneous subtype of BCA [71,72].

## 5. Immunotherapy for BCA

Neoplastic cells elude immune surveillance through a myriad of strategies, further establishing treatment resistance by fostering an immunosuppressive TME. This environment is characterized by the upregulation of immune checkpoint inhibitors (ICIs), including *PD-L1/2*, *CTLA-4*, *TIM-3*, *LAG-3*, *BTLA*, *ITIM*, and *TIGIT*, alongside the recruitment of immunosuppressive cell populations, such as M2 macrophages, regulatory T-cells (Tregs), and myeloid-derived suppressor cells (MDSCs) [73,74,75]. The elucidation of these tumor evasion mechanisms has paved the way for the development of innovative therapeutic modalities termed “immunotherapy”. This approach seeks to dismantle tumor-induced immunosuppression, leveraging the immune system’s inherent specificity and cytotoxic potential to target and eradicate cancer cells [22]. Historically, BCA tumors were deemed non-immunogenic or immunologically quiescent, rendering them unsuitable for immunotherapeutic interventions [76,77]. Subsequent research, however, has revealed significant levels of tumor-infiltrating lymphocytes (TILs) within more aggressive BCA subtypes, such as TNBC and Her-2 positive, correlating with enhanced 5-year overall survival (OS; 74.3% in patients with high TILs vs. 52.0% in those with low TILs), pathologic complete response (pCR; 40–50% in the high TIL group), and disease-free survival (DFS) [78,79,80]. These findings further identified diminished TIL activity and PD-1 expression in hormone receptor-positive (HR+) BCA subtypes, suggesting subtype-specific immune microenvironments. The KEYNOTE trials pioneered the assessment of immunotherapy’s safety and efficacy in metastatic TNBC, with the KEYNOTE-355 trial demonstrating that metastatic TNBC patients exhibiting high PD-L1 expression and treated with Pembrolizumab in combination with chemotherapy exhibited improved progression-free survival (PFS) and OS [81]. Similarly, the IMpassion130 trial reported clinically meaningful OS benefits in TNBC patients receiving first-line atezolizumab plus nab-paclitaxel treatment, with a hazard ratio (HR) of 0.62; a 95% confidence interval (CI), 0.49–0.78; and *p* < 0.0001 [82]. The limited clinical benefit of ICIs in most BCA patients emphasizes the need for more effective immunotherapy, such as cellular therapy.

Chimeric antigen receptor (CAR) T-cell therapy represents an advanced form of cellular immunotherapy, originating from the concept of adoptive T-cell transfer. This therapy involves the genetic modification of T-cells to express chimeric receptors that comprise an antigen-specific single-chain variable fragment (scFv) coupled with costimulatory domains. This innovative approach capitalizes on the patient’s autologous immune cells to target and eradicate cancerous cells [22,76]. Clinical investigations predominantly target disseminated tumor cells with the capacity to remain quiescent for extended periods, yet possess metastatic potential, particularly in the context of BCA. A significant focus of these clinical trials is on TNBC, characterized by its resistance to conventional therapies due to the absence of ER, PR, and Her-2 receptors. This resistance underscores the necessity for novel therapeutic strategies, rendering CAR T-cell therapy a promising option for TNBC treatment [83]. Mucin1 (MUC1) has been identified as a tumor-associated antigen (TAA) that is intricately linked with the invasive and metastatic capabilities of cancerous cells across a variety of cancers, including TNBC [84,85]. A specific aberrant glycoform of MUC1, which is present in approximately 95% of cancer cells, serves as a target for the development of CAR T-cell therapies, referred to as tMUC1-targeted CAR T-cells [86]. Wallstabe L et al. (2018) have identified Integrin αvβ3 as another viable TAA for the creation of CAR T-cell therapies aimed at BCA. Targeted destruction of αvβ3 via αvβ3-CAR T-cells has been proposed as a strategy to inhibit angiogenesis within the TME, potentially limiting tumor growth and metastasis [87]. Numerous investigations have identified a range of TAAs as potential targets for the development of CAR T-cell therapies aimed at treating Her-2-positive and TNBC subtypes [49,88,89]. However, the utility of these TAAs is often limited by their heterogeneous expression and the phenomenon of antigen escape, which can undermine the efficacy of CAR T-cell immunotherapy in these aggressive cancer subtypes. Therefore, the selection of a suitable target antigen, alongside the implementation of advanced genetic engineering techniques to shield CAR T-cells from the immunosuppressive effects of the TME, is paramount for optimizing therapeutic outcomes [90]. In addressing the challenge of heterogeneous TAA expression, the development of advanced CAR T-cells capable of dual antigen targeting represents a promising strategy for enhancing the precision and effectiveness of treatments for Her-2-positive and TNBC tumors. Our review highlights IL-13Rα2 and EphA2 as clinically relevant biomarkers for these cancer subtypes, noting their minimal expression in normal tissue, which underscores their potential as targets for CAR T-cell therapy.

CAR T-cell therapy emerges as a compelling therapeutic strategy, especially for patients with Her-2 enriched and TNBC, who typically face high metastasis rates, early recurrence, and poorer prognoses [44,45,91,92]. The receptors EphA-2 and IL-13Rα2 are notably overexpressed on tumor cells in these subtypes compared to normal epithelial cells, suggesting their potential as targets for CAR T-cell therapy [3,13,16,29,33,93,94,95,96,97]. In contrast, these receptors exhibit basal or negligible expression in Luminal BCA subtypes, akin to normal epithelial cells, highlighting their specificity for aggressive cancer forms. Numerous studies have investigated the expression and clinical relevance of EphA-2 and IL-13Rα2 in BCA. For instance, a study underscored the clinical significance of EphA2, synthesizing data from preclinical studies utilizing animal models or BCA cell lines [98]. Immunohistochemical analyses have identified EphA2 expression in Her-2 and TNBC subtypes, establishing its correlation with clinical parameters, such as tumor grade, stage, and patient survival [8]. EphA-2’s involvement in resistance mechanisms against trastuzumab therapy has also been documented, further validating its role as a key pro-tumorigenic antigen within the breast cancer landscape [38]. Furthermore, EphA-2 is a more prominent pro-tumorigenic antigen in breast cancer than other Eph family members [10,99]. Therapeutic approaches targeting EphA-2, including small molecular kinase inhibitors, antibody–drug conjugates, and ephrin A1 mimetic peptides, have demonstrated promising therapeutic potential, reinforcing the utility of EphA-2 and IL-13Rα2 as valuable biomarkers for the development of targeted CAR T-cell therapies in aggressive BCA subtypes [99].

The differential expression of IL-13Rα2 in normal versus tumor tissue, coupled with the receptor’s role in facilitating tumor growth, survival, invasion, and metastasis through IL-13Rα2-mediated signaling, underscores its potential as a therapeutic target [3,13,16,29]. This variability in expression profiles, along with the capacity of IL-13Rα2 to influence tumor cell survival and proliferation via receptor signaling mechanisms, renders IL-13Rα2 and EphA-2 promising targets for cancer therapy. Consequently, EphA-2 and IL-13Rα2 emerge as potential target antigens for CAR T-cell therapy, particularly in the context of Her-2-enriched and TNBC subtypes, where their overexpression could be leveraged for targeted therapeutic interventions.

## 6. IL-13R Alpha-2 and EphA2 as Therapeutic Targets

The EphA2 receptor, as the inaugural member of the Eph receptor family, is increasingly recognized as a viable therapeutic target for oncological interventions. A variety of molecular entities, including monoclonal antibody (mAb)–drug conjugates and small molecule inhibitors, have been engineered to inhibit the kinase activity of EphA2. These therapeutic agents function by obstructing the interaction between the ephrin A1 ligand and the EphA2 receptor. For example, in the context of MDA-MB-231 breast cancer cell lines, the engagement of the EA1.2 monoclonal antibody with the extracellular domain of EphA2 initiates receptor phosphorylation, culminating in its subsequent degradation [100]. Bruckheimer EM et al. (2009) employed the monoclonal antibody 3F2-3M as a strategy to attenuate the proliferation of MDA-MB-231 breast cancer cells through the induction of antibody-dependent cellular cytotoxicity (ADCC) activity [101]. Additionally, Gokmen-Polar et al. (2011) utilized the anti-EphA2 antibody EA5 to target Luminal BCA cells, successfully mitigating Tamoxifen resistance. This approach underscores the potential of EphA2-targeted therapies in overcoming resistance mechanisms in BCA treatment paradigms [102]. The pharmacological agent ALW-II-41-27, categorized as a tyrosine kinase inhibitor, has been applied in clinical settings for the treatment of triple-negative/basal-like BCA tumors exhibiting EphA2 positivity. This compound effectively attenuates the tyrosine kinase activity of EphA2 through the inhibition of Y588 phosphorylation, highlighting its potential as a targeted therapeutic strategy against specific BCA subtypes [34]. The WW437, classified as a histone deacetylase inhibitor, has demonstrated a substantial decrease in EphA2 expression, subsequently leading to a reduction in both the proliferation and metastasis of BCA cells. This effect underscores the potential utility of WW437 in targeting the molecular pathways associated with BCA progression and dissemination [103]. Utilizing a combinatorial therapeutic strategy, stealth liposomes modified with an EphA2-binding homing peptide (YSA-LP) in conjunction with Doxorubicin (DOX) have demonstrated enhanced efficacy against cancer cells. This approach capitalizes on the specificity of YSA-LP for EphA2-expressing cells to facilitate targeted delivery of DOX, thereby improving the therapeutic impact on cancer cells through increased precision and potency [94]. In parallel, the pharmacological compound “MSN-YSA”, when co-administered with Doxorubicin, exhibited in vitro antitumor activity against BCA cell lines. This observation underscores the potential utility of MSN-YSA as a targeted therapeutic agent, enhancing the cytotoxic efficacy of Doxorubicin through a synergistic mechanism against malignant cells in BCA research models [104]. Salem AF and colleagues (2018) demonstrated the successful eradication of circulating BCA cells utilizing the “123B9” peptide molecule in conjunction with Paclitaxel (PTX). This combination therapy highlights a significant advance in targeting metastatic cells within the bloodstream, offering a promising strategy for the treatment of BCA through the synergistic effects of the peptide molecule and chemotherapeutic agent [105]. In the EMT6 breast tumor model, the deployment of EphA2-targeted liposomal paclitaxel, termed “EphA2-il-dtxp”, in conjunction with the PD-L1 blockade, elicited a significant tumor response in vivo. This combination therapy underscores the potential of targeted delivery systems in enhancing the efficacy of immunotherapeutic agents and chemotherapy, thereby providing a promising approach for the treatment of BCA through synergistic mechanisms of action [106]. Table 1 summarizes clinical studies on targeting EphA2 in breast cancer.

The targeting of IL-13Rα2 has been identified as a viable strategy for anticancer therapy, paralleling the promising targeting of EphA1. The IL-13Rα2 DNA vaccine, in combination with IL13-PE (a chimeric pseudomonas exotoxin), synergized to reduce murine breast tumor growth by multiple mechanisms, including direct tumor killing and increased T-cell tumor infiltration [107]. Investigations have focused on a hybrid cytolytic peptide, comprising a unique receptor-binding domain (Pep-1) and a cytolytic domain (Phor21), tested against gliomas expressing IL-13Rα2. This approach induced apoptosis in cancer cells, demonstrating the potential of receptor-specific peptides in mediating targeted cytotoxicity against malignancies expressing IL-13Rα2 [16]. In the context of preclinical and Phase I to III clinical trials, researchers have developed a truncated recombinant immunotoxin derived from “Pseudomonas aeruginosa exotoxin A” conjugated with IL-13 (PE-IL-13) aimed at targeting IL-13Rα2. The outcomes of these investigations have demonstrated significant cytotoxic effects on cancer cell populations in GBM, renal cell carcinoma (RCC), and head and neck squamous cell carcinoma (HNSCC), highlighting the potential of this targeted approach in the treatment of cancers expressing IL-13Rα2 [108]. Researchers have characterized two variants of IL-13Rα2, designated “C4” and “D7”, for their utility as binding domains within CAR T-cells. These studies aimed to elucidate the capacity of these variants to specifically target and elicit responses against IL-13Rα2-expressing cells, thereby contributing to the development of precision immunotherapy strategies targeting this receptor [109]. Pandya H et al. (2012) elucidated the selective binding affinity of the peptide “Pep-1L” for IL-13Rα2, identifying a non-competitive binding domain distinct from the IL-13 interaction site within a GBM xenograft model. This investigation revealed that the Pep-1L peptide exhibits specific binding to IL-13Rα2, utilizing a binding mechanism that does not compete with the IL-13 receptor interaction [110]. Bartolome et al. (2018) developed a 12-mer peptide, designated D1, which selectively inhibits the IL-13/IL-13Rα2 signaling pathway. This intervention resulted in the attenuation of cell migration and invasion across CRC and GBM cell lines, alongside an enhancement in the survival rates of xenograft models [111]. Jaen et al. (2021) employed a similar strategy to synthesize a monoclonal antibody (mAb) specific to the D1 peptide, targeting the IL-13/IL-13Rα2-mediated signaling pathway. This D1-specific mAb effectively diminished liver metastasis CRC tumors and subsequently improved survival outcomes [27]. Balyasnikova et al. also developed a new antibody to inhibit interaction between IL-13/IL-13Rα2 in the orthotopic GBM xenografts model [112]. Clinical trials have explored the use of peptide-pulsed dendritic cells (DCs) as a cancer vaccine for advanced glioma patients, targeting IL-13Rα2. These studies have documented T-cell responses to the peptide-pulsed DCs [113,114]. Despite various targeted strategies against EphA-2 and IL-13Rα2 to curb tumor growth, the efficacy of these interventions remains constrained. Consequently, a combinatorial therapeutic approach targeting both EphA-2 and IL-13Rα2 concurrently may represent a more effective strategy. Figure 4 shows different immunotherapeutic targeted approaches for IL-13Rα2 and EphA2.

## 7. IL-13Rα-2 and EphA2 as CAR T-Cell Targets

IL-13Rα2’s rarity in normal tissues positions it as an optimal target for antigen-specific therapies. The efficacy of IL-13Rα2-targeted CAR T-cell therapy, initially explored in GBM, is now being extended to other malignancies, including BCA. Current clinical trials, many of which have reported promising outcomes, are exploring various CAR T-cell constructs targeting IL-13Rα2 (clinicaltrials.gov). These constructs vary in their design, featuring distinct hinge regions, cytoplasmic domains for cytotoxic signaling, and mechanisms for antigen recognition. Based on the evidence reviewed, our laboratory is developing a dual antigen-targeting CAR T-cell therapy that targets both EphA-2 and IL-13Rα2 for the treatment of Her-2 enriched and TNBC, although specific data have not been disclosed. We posit that employing dual antigen-targeting strategies, such as TanCAR, Biphasic CAR, and SynNotch CAR, which simultaneously target two antigens, could offer therapeutic advantages. This hypothesis is supported by studies in other cancer models, including an orthotopic GBM model, where dual-targeted Her-2/IL-13Rα2 CAR T-cells demonstrated superior tumor suppression and survival enhancement compared to treatments with monospecific CAR T-cells, whether applied solo or in combination [115].

Subsequent investigations have revealed that, in comparison to monospecific CAR T-cells, a bispecific TanCAR configuration incorporating a “Her-2-binding single-chain variable fragment (scFv)/mutated IL-13” heterodimer has demonstrated enhanced survival rates in GBM models. This bispecific approach allows for the simultaneous targeting of two distinct tumor-associated antigens, potentially increasing the efficacy of CAR T-cell therapy against malignant cells [116]. An engineered trivalent CAR T-cell through the introduction of a singular universal tricistronic transgene enabled the concurrent expression of CARs targeting Her-2, IL-13Rα2, and EphA2. This innovative approach aimed to augment the therapeutic potential of TanCAR T-cells. In the treatment of GBM xenograft models, the application of these trivalent TanCAR T-cells resulted in prolonged survival times and a reduction in tumor burden, demonstrating an enhanced patient-matched T-cell response against the malignancy [117]. Despite the therapeutic advantages of CAR T-cell therapy, several challenges remain to be addressed, including associated toxicities, limited efficacy in the context of solid tumors, antigen escape phenomena, reduced persistence, suboptimal trafficking or tumor penetration, and the counteractive effects of the immunosuppressive TME [118,119,120]. Numerous strategies have been proposed to enhance the antitumor efficacy and mitigate the toxicities associated with cellular therapy in solid tumors [20]. For instance, the deployment of bispecific CARs capable of recognizing multiple antigens may circumvent the challenge of antigen loss by employing a combination of CAR T-cells, each targeting a distinct antigen. Notably, a cohort of cholangiocarcinoma patients demonstrated enhanced therapeutic outcomes following treatment with a composite of CAR T-cells specific to epidermal growth factor receptor (EGFR) and CD133 [121]. In a non-small cell lung cancer (NSCLC) model, a therapeutic combination of CAR T-cells targeting prostate stem cell antigen (PSCA) and mucin 1 (MUC1) effectively eradicated cancer cells expressing PSCA and MUC1 [122]. Furthermore, chemokines originating from tumors represent a promising avenue for the recruitment of tumor-reactive T-cells. Specifically, CAR T-cells modified to express the chemokine receptor CXCR2, which interfaces with the ligand CXCL1 present on melanoma cells, exhibited proficient migration toward the tumor microenvironment [123]. CXCL1 is chemotactic of CXCR2 expressing CAR T-cells toward the tumor [122].

In response to insufficient or aberrant vascularization, research endeavors have aimed at vascular stroma through the utilization of anti-angiogenic compounds. Wang et al. elucidated that Vascular Endothelial Growth Factor Receptor-1 (VEGFR-1) CAR T-cells attenuate resistance to traditional treatments directed at angiogenesis, concurrently enhancing the tumor-killing capacity of CAR T-cells [124]. Targeting immunosuppressive cells within the TME, including regulatory T-cells (Tregs), myeloid-derived suppressor cells (MDSCs), and M2 macrophages, enhances the effectiveness of CAR T-cell therapy. The modification of CAR T-cells to express cytokines, such as interleukin-12 (IL-12), IL-18, and IL-15, facilitates the modulation of the immunosuppressive TME, thereby promoting enhanced survival of CAR T-cells and recruitment of T-memory cells and central-memory T-cells [125].

In the setting of monotherapy directed at a single antigen, a decline in its expression often occurs, likely attributed to selective elimination and consequent tumor evasion as a resistance mechanism. Consequently, employing a combinatorial targeting approach, such as IL-13Rα2- and EphA2-targeted therapy, which addresses multiple distinct antigens, may offer enhanced efficacy. This multifaceted targeting strategy has the potential to mitigate tumor antigenic heterogeneity and deter antigen evasion. Nevertheless, the pursuit of multiple antigens as therapeutic targets may engender challenges, such as “on-target, off-tumor” and “off-target toxicities”, particularly neurotoxicity in solid tumors. To circumvent these issues, innovative strategies have been devised, including the integration of suicide genes, targeting two tumor-associated antigens (TAAs), switch-mediated activation, and other cutting-edge gene therapy methodologies [126,127]. Moreover, the immunosuppressive milieu within TIME significantly influences the efficacy of immunotherapy, posing a significant challenge to CAR T-cell effectiveness. Addressing this issue entails the manipulation of the intricate interplay among immunosuppressive constituents. This can be accomplished by targeting immune suppressive cells, such as MDSCs, tumor-associated macrophages (TAMs), Tregs, regulatory B-cells, and tumor-associated neutrophils. Additionally, the modulation of the cytokine and chemokine milieu, immune checkpoint pathways, and tumor antigen heterogeneity represents potential strategies to mitigate this immunosuppressive landscape [128] (Figure 5).

In recent years, there has been growing interest in the development of CAR T-cell therapies capable of targeting two tumor antigens simultaneously, known as “dual targeting CAR” approaches (Figure 6). This involves engineering T-cells to express two distinct chimeric antigen receptors, each targeting a different TAA present on tumor cells. Various strategies and formats for these dual-targeting CARs have been explored, including cocktail infusion of separate single-target CAR T-cell products, heterogeneous cell products combining single- and bi-target CAR T-cells, bicistronic bi-target CARs, tandem bi-target CARs, and loop bi-target CARs. Such approaches not only mitigate immunotherapy-related toxicities but also demonstrate improved OS. For instance, dual-targeting CD19/CD22 CAR T-cell therapy has shown enhanced OS and progression-free survival (PFS) in Acute Myeloid Leukemia (AML) patients compared to single-antigen CAR T-cell therapy [129,130,131,132,133] (Figure 7). Moreover, dual-targeting approaches are associated with a reduced risk of severe cytokine release syndrome and neurotoxicity, as well as limiting tumor antigen escape mechanisms [131,132,133].

Numerous investigations have provided evidence supporting the notion of employing combination therapy as a strategy to augment the efficacy of CAR T-cell therapy [134,135]. In this context, chemotherapy, radiotherapy, immune checkpoint inhibitors (ICIs), cytokines, immunomodulatory agents, cancer vaccines, oncolytic viruses, hematopoietic stem cell transplantation (HSCT), and metabolic inhibitors represent exemplary therapeutic modalities combined with CAR T-cell therapy. Integration of these combinatorial treatment regimens serves to augment the safety and efficacy profile of CAR T-cell therapy while mitigating associated toxicities.

**Figure 6 ijms-25-03780-f006:**
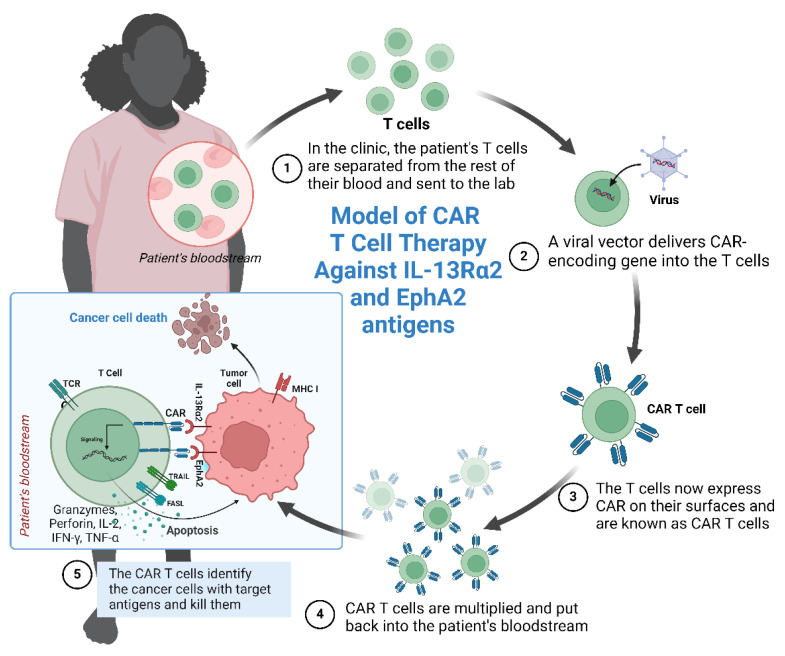
A CAR T-cell designed for targeting IL-13Rα2 and EphA2 antigens on breast cancer cells (Created with BioRender.com).

**Figure 7 ijms-25-03780-f007:**
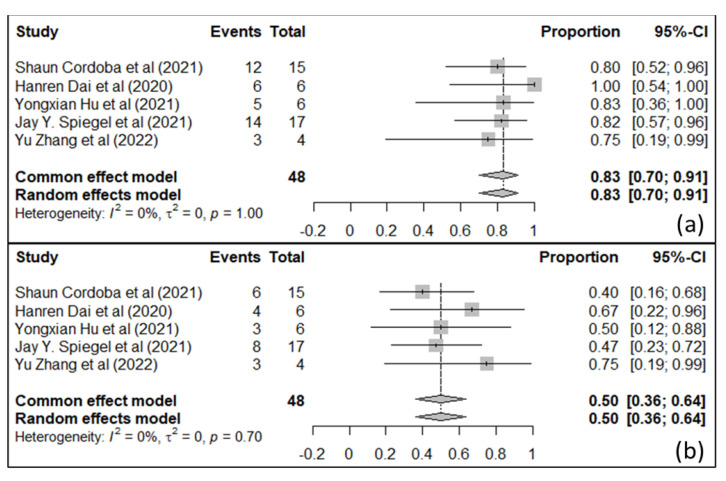
Forest plot for survival outcome in six months for acute lymphoid leukemia [131,132,136,137,138]: (**a**) six-month overall survival, (**b**) six-month event-free survival (Reprinted/adapted with permission from Ref. [139]. Copyright © 2023 The Authors. Cancer Medicine published by John Wiley & Sons Ltd.).

The present review highlights the oncogenic significance of IL-13Rα2 and EphA2 in highly aggressive BCA subtypes, including Her-2 enriched and TNBC. EphA2 and its ligands play crucial roles in the development, metastasis, and initiation of human BCA, with EphA2 overexpression being particularly prevalent in TNBC and Her-2-enriched subtypes and associated with poor prognosis [34,140,141]. Both IL-13Rα2 and EphA2 have been implicated in promoting the growth, survival, migration, and invasion of BCA cells in in vitro and in vivo models. Consequently, they represent promising therapeutic targets for BCA treatment. Various targeted regimens, such as peptides, antibodies, small molecule inhibitors, and antibody–drug conjugates (ADCs), have shown significant anticancer effects in IL-13Rα2- and EphA2-positive BCA models, either as monotherapies or in combination with other treatments [140,141].

In conclusion, IL-13Rα-2 and EphA2 emerge as highly promising therapeutic targets for Her-2-enriched and TNBC subtypes of BCA. However, the effectiveness of CAR T-cell therapy in solid tumors is limited by several challenges, including toxicity to normal tissues, limited efficacy, antigen escape, poor persistence, difficulties in trafficking, and an immunosuppressive TME.

Addressing these issues necessitates further research into the functional mechanisms of IL-13Rα2 and EphA2, their interacting proteins, and upstream regulatory mechanisms, to inform the development of targeted therapies. Consequently, clinical research efforts should focus on developing cellular therapies utilizing IL-13Rα2 and EphA2 as antigens for the treatment of Her-2-enriched and TNBC subtypes, which are characterized by high aggressiveness and associated mortality rates.

Importantly, breast cancer manifests as a heterogeneous condition. The expression levels of IL-13Rα2 and EphA2 may exhibit variability among patients with HER-2 enriched and triple-negative breast cancer (TNBC). Thus, it is imperative to assess antigen expression using quantitative techniques, such as immunohistochemistry or real-time PCR, promptly before initiating therapy.

## 8. Conclusions

In conclusion, the exploration of IL-13Rα2 and EphA2 within the context of BCA has highlighted their pivotal roles in tumor progression, metastasis, and the development of resistance mechanisms. We discussed the aberrant overexpression of these receptors in the more aggressive subtypes of BCA, such as Her-2 enriched and TNBC, associating their presence with poorer prognostic outcomes. IL-13Rα2 and EphA2 structural and functional characteristics, alongside their pathological significance in BCA, present them as potential therapeutic targets.

We reviewed the challenges inherent in the current therapeutic landscape for BCA, especially in treating Her-2-enriched and TNBC subtypes, which exhibit high rates of metastasis, recurrence, and mortality. Evidence suggests that the targeting of IL-13Rα2 and EphA2 through innovative immunotherapeutic strategies, such as chimeric antigen receptor T-cell (CAR-T) therapy, is a promising avenue for overcoming these hurdles. Preclinical and clinical studies demonstrate the feasibility and therapeutic potential of IL-13Rα2 and EphA2 as biomarkers for the development of CAR T-cell therapies tailored to combat these aggressive forms of BCA.

There is a necessity to address the challenges that CAR T-cell therapy faces, particularly in solid tumors, including issues related to toxicity, efficacy, antigen escape, and the immunosuppressive tumor microenvironment (TME). There is also a need for ongoing research into the mechanisms underlying the roles of IL-13Rα2 and EphA2 in cancer progression, the development of resistance, and the interplay with the TME.

Ultimately, the targeted manipulation of IL-13Rα2 and EphA2 presents a significant opportunity for advancing the treatment of Her-2-enriched and TNBC subtypes. By leveraging the unique properties of these receptors, it is possible to develop more effective, targeted therapies that can improve outcomes for patients with these challenging forms of breast cancer.

## Figures and Tables

**Figure 1 ijms-25-03780-f001:**
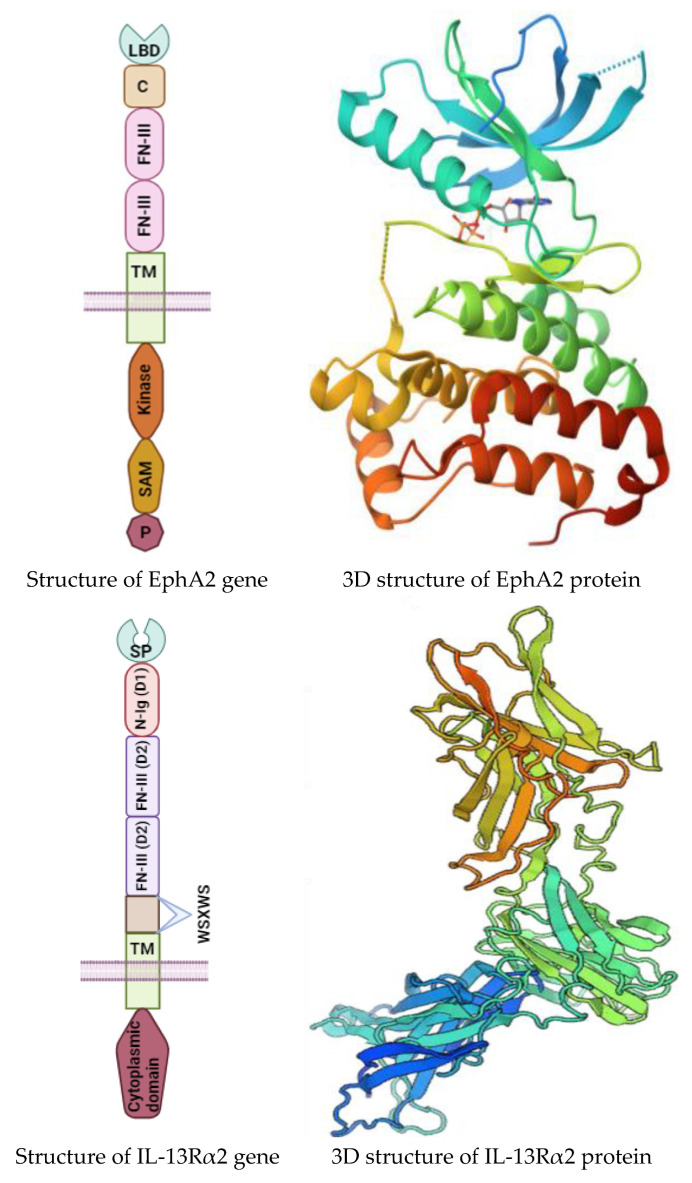
Structure of IL-13Rα2 and EphAR2 genes (Created with BioRender.com) and their protein 3D structures.

**Figure 2 ijms-25-03780-f002:**
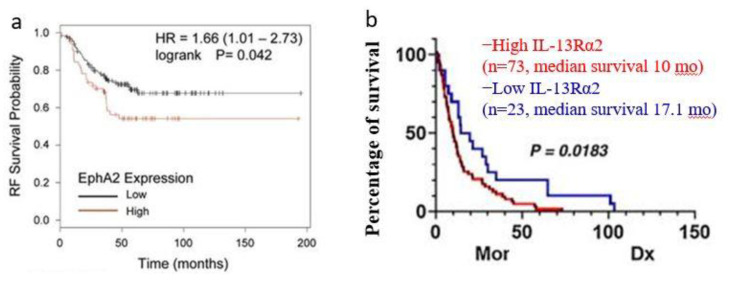
Survival curve analysis shows that a higher expression of EphA2 (**a**) and IL-13Rα2 (**b**) correlates with a lesser survival of Her-2-enriched and TNBC patients (Reprinted/adapted with permission from Refs. [3,34]. Copyright ©2021 The Authors; Published by the American Association for Cancer Research and Copyright © 2017 The Author(s)).

**Figure 3 ijms-25-03780-f003:**
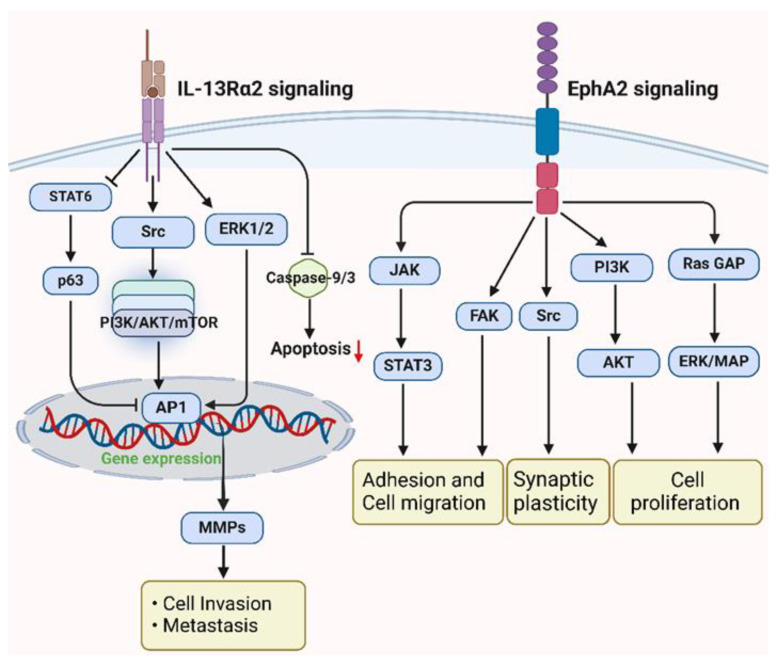
Illustration of IL-13Rα2- and EphA2-mediated tumorigenic signaling in breast cancer. Both activated pathways regulate cell adhesion, invasion, metastasis, and cell plasticity via regulating MMPs, PI3K/AKT/mTOR, and ERK/MAP signaling (Created with BioRender.com).

**Figure 4 ijms-25-03780-f004:**
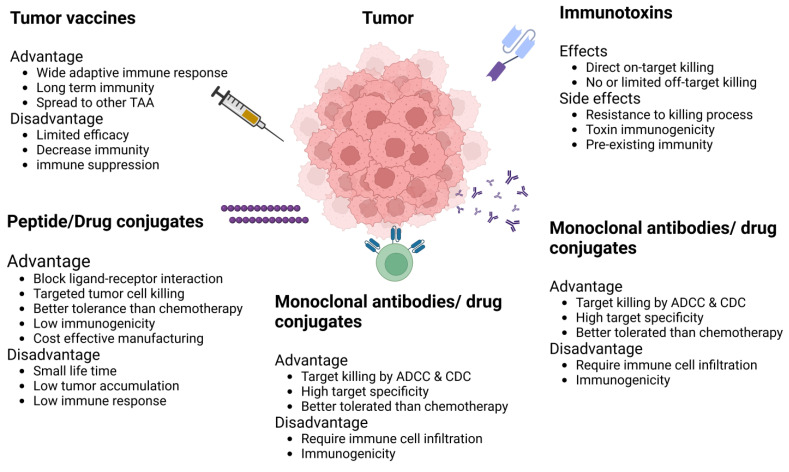
Illustration of different immunotherapeutic approaches to targeted IL-13Rα2 and EphA2.

**Figure 5 ijms-25-03780-f005:**
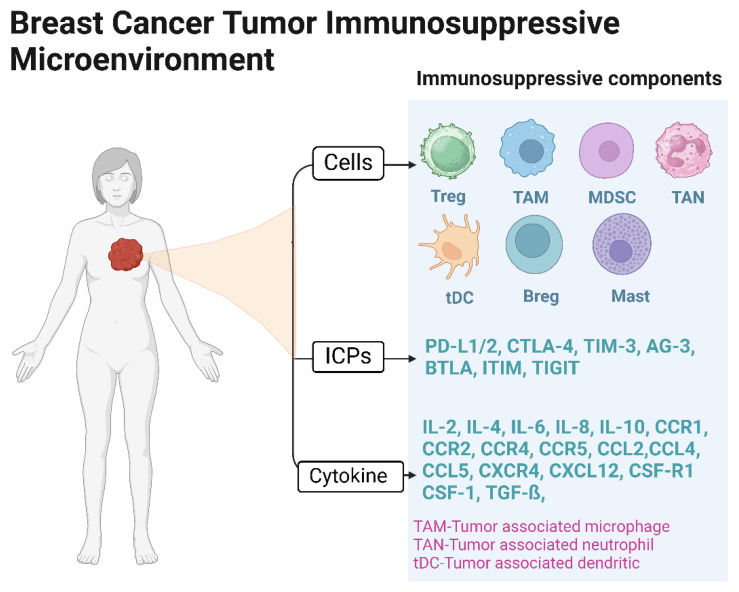
Breast cancer immunosuppressive components, such as immune suppressive cells (MDSCs, tumor-associated macrophages, Tegs and regulatory B-cells, and tumor-associated neutrophils), cytokine and/or chemokine milieu, and immune checkpoint proteins.

**Table 1 ijms-25-03780-t001:** Clinical studies on targeting EphA2 in breast cancer.

Study Compound	Agent Type	Phase	Trial Identifier	Status
DS-8895a	Monoclonal antibody, immunotherapy	1	NCT02252211	Completed
DOPC-liposomal EphA2 siRNA	Silencing RNA/RNAi, nanotechnology	1	NCT01591356	Recruiting
Dasatinib + Zoledronic Acid	Small-molecule inhibitor	1/2	NCT00566618	Completed
Dasatinib	Small-molecule inhibitor	2	CA180059	Completed
Dasatinib	Small-molecule inhibitor	2	NCT02720185	Recruiting
Various agents, including Dasatinib and Ponatinib	Precision medicine	1b	NCT03878524	Recruiting
Various agents, including Dasatinib	Precision medicine	2	NCT02465060	Recruiting
BT5528	Small-molecule inhibitor	1/2	NCT04180371	Recruiting

## Data Availability

Not applicable.

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
