# Peer review of "Targeting Interleukin-13 Receptor α2 and EphA2 in Aggressive Breast Cancer Subtypes with Special References to Chimeric Antigen Receptor T-Cell Therapy"

_ijms, 2024, doi:10.3390/ijms25073780_

Round 1

Reviewer 1 Report

Comments and Suggestions for Authors

The authors discuss the therapeutic potential of targeting interleukin-13 receptor α2 and erythropoietin-producing hepatocellular receptor A2 in breast cancer, particularly in aggressive subtypes such as Her-2 enriched and triple-negative breast cancer (TNBC). The authors highlight the overexpression of these receptors in breast cancer compared to normal tissues and their roles in tumor progression and resistance. They examine the structural and functional aspects of EphA2 and IL-13Rα2, their pathological significance in breast cancer, and the promising therapeutic avenues their targeting presents. The article also delves into the current immunotherapeutic strategies, including the limitations of existing treatments and the potential of dual antigen-targeting chimeric antigen receptor T-cell (CAR-T) therapies. The authors emphasize the urgent need for targeted therapies to combat the high mortality rates associated with Her-2 enriched and TNBC subtypes. They discuss the potential role of EphA2 and IL-13Rα2 as promising candidates for the development of CAR-T cell therapies, based on a thorough examination of preclinical and clinical studies.  

Comments

  1. While the article discusses preclinical and clinical studies, there seems to be a lack of extensive clinical data on the efficacy of targeting EphA2 and IL-13Rα2 in breast cancer patients.
  2. Breast cancer is a highly heterogeneous disease, and the expression of EphA2 and IL-13Rα2 may vary among patients. The study should address this heterogeneity and develop strategies to identify patients who are most likely to benefit from the targeted therapies.
  3. Targeting EphA2 and IL-13Rα2 may have off-target effects on normal tissues expressing these receptors, leading to toxicity. The study should investigate ways to minimize off-target effects and enhance the specificity of the targeted therapies.
  4. The immunosuppressive tumor microenvironment poses a significant challenge for CAR-T cell therapies. The study should explore strategies to overcome this barrier, such as combining CAR-T cells with immune checkpoint inhibitors or other immunomodulatory agents.
  5. Tumor cells may downregulate or lose the expression of targeted antigens, leading to antigen escape and treatment resistance. The study should address this issue by developing multi-antigen targeting approaches or identifying alternative targets.
  6. The potential synergistic effects of combining targeted therapies with other treatment modalities, such as chemotherapy or radiotherapy, should be explored to enhance the overall therapeutic efficacy.
  7. The study should investigate the potential mechanisms of resistance to EphA2 and IL-13Rα2 targeted therapies and develop strategies to overcome them.

Reviewer 2 Report

Comments and Suggestions for Authors

The manuscript titled "Targeting Interleukin-13 Receptor α2 and EphA2 in Breast Cancer: With special references to CAR T-cell therapy" presents a review of the roles of interleukin-13 receptor α2 (IL-13Rα2) and erythropoietin-producing hepatocellular receptor A2 (EphA2) in breast cancer, especially focusing on their implications for chimeric antigen receptor T-cell (CAR-T) therapy. The manuscript does a good job of summarizing the existing literature on IL-13Rα2 and EphA2, their roles in breast cancer progression, and the potential for targeting these receptors in CAR-T cell therapy. The inclusion of a wide range of studies, including preclinical and clinical investigations, provides a solid foundation for the review. Novelty exists in this manuscript, but apparent flaws can also be observed. Thus, major revision is recommended. Main issues have been listed as followed:

1.More discussion on emerging research areas, potential strategies to overcome resistance mechanisms, and the latest advancements in CAR-T cell design and application, are recommended.

2.While the focus on CAR-T cell therapy is valuable, a comparison with other immunotherapeutic approaches targeting IL-13Rα2 and EphA2 could provide a more comprehensive overview of the therapeutic landscape. This could include discussion on monoclonal antibodies, small molecule inhibitors, and combination therapies.

3.A section of limitations is recommended to summarize and discuss the current related studies and the gaps in knowledge, such as the potential side effects, the heterogeneity of breast cancer subtypes, and the impact of the tumor microenvironment on therapy effectiveness.

4.Relevant studies (including clinical and laboratory research) should be detailed and listed in a table.

5.Careful proofreading is required to correct typographical errors and ensure consistency in formatting, particularly in the references section.

6. More tables and figures are required.

Reviewer 3 Report

Comments and Suggestions for Authors

This manuscript, written by Dr Bir, review type, with the title of "Targeting Interleukin-13 Receptor α2 and EphA2 in Breast Cancer: With special references to CAR T-cell therapy" reviews and summarizes relevant information about
EphA2 and IL-13Rα2. The text is well written, it is easy to understand. However, some sections are vary large; and may benefit from transforming the data from the text to a table adn/or figure.

(1) Line 30-31. Regarding "Breast cancer (BCA) represents the foremost cause of cancer-related mortality among 30 women, accounting for 31% (297,790 cases) of all cancer diagnoses in 2023".
could you please add the frequencies of the 2nd, and 4th cause? What about colorectal cancer?

(2) Line 71. Regarding "Tumorigenic mechanisms of action of EphA2 and IL-13R2α". Is this title correct? Looks more like types of therapy against tumors.

(3) Line 109. Regarding "Targeted strategies for IL-13R2α and EphAR2". Looks like the title is also incorrect, as this figure is showing the structure of IL13R2a and EphA2.

(4) Regarding Figure 2. Is it possible to show the 3D structure of IL-13R2α and EphAR2?

(5) Line 112. Could you please descrive what is and why triple negative breast cancer is important?

(6) Line 116. Is it correct to cite proceedings data (withouth peer review). Has this abstract been published as regular article?

(7) Line 121. Please descrive briefly the Nottingham prognostic index.
https://www.ncbi.nlm.nih.gov/pmc/articles/PMC3151231/

(8) Lines 123-136. Is it possible to depict the mechanim in a figure?

(9) Lines 155-156. Regarding "These subtypes exhibit unique gene ex-155 pression profiles, dictating divergent clinical outcomes". Is there any available for publishing surival plot?

(10) Line 187. Please write gene names in italics. Please revise all the text.

(11) Line 193. Please make a figure with the different immune checkpoit molecules, and please expand the immune microenvironment as well.

(12) Line 383. Please show a figure with the CAR-T gherapy.

(13) Lines 405-409. How CAR-T cells express these cytokines and how the cytokines modify the microenvironment. Please add more details.

(14) Lines 423-425. Please add the figure of surival analysis, or the hazard risk (with p values).

(15) Title mentions breast cancer, but in abstract the subtype of TNBC wa chosen. Why not changing the title?

(16) Are these two markers expressed in all tissues and/or neoplastic transformation?

(17) Apart from breast cancer, what about othe relevant cancer locations/subtypes?

Round 2

Reviewer 2 Report

Comments and Suggestions for Authors

Relevant studies (including clinical and laboratory research) have been added and listed in tables. More tables and figures have also been added in this revision. The revised manuscript provides us a more thorough review than before, demonstrating a deeper understanding of the current state of breast cancer research, the biological roles of IL-13Rα2 and EphA2 in cancer progression, and the development and challenges of CAR T-cell therapy. The manuscript indeed, presents novel insights into the potential mechanisms by which IL-13Rα2 and EphA2 contribute to the aggressiveness of certain breast cancer subtypes. The hypotheses proposed regarding the utility of these receptors as targets for CAR T-cell therapy are compelling and warrant further investigation. Acceptance is recommended.